# Improving the Efficiency of Viability-qPCR with Lactic Acid Enhancer for the Selective Detection of Live Pathogens in Foods

**DOI:** 10.3390/foods13071021

**Published:** 2024-03-27

**Authors:** Laura-Dorina Dinu, Quthama Jasim Al-Zaidi, Adelina Georgiana Matache, Florentina Matei

**Affiliations:** 1Faculty of Biotechnology, University of Agricultural Sciences and Veterinary Medicine, 011464 Bucharest, Romania; alzaidy2004.n3@gmail.com (Q.J.A.-Z.); adelinamatache21@gmail.com (A.G.M.); florentina.matei@biotehnologii.usamv.ro (F.M.); 2Faculty of Food Industry and Tourism, Transilvania University of Brasov, 500015 Brasov, Romania

**Keywords:** quantitative PCR, propidium monoazide (PMA), lactic acid, detection of viable pathogens, food matrix

## Abstract

Pathogenic *Escherichia coli* are the most prevalent foodborne bacteria, and their accurate detection in food samples is critical for ensuring food safety. Therefore, a quick technique named viability-qPCR (v-qPCR), which is based on the ability of a selective dye, such as propidium monoazide (PMA), to differentiate between alive and dead cells, has been developed. Despite diverse, successful applications, v-qPCR is impaired by some practical limitations, including the ability of PMA to penetrate the outer membrane of dead Gram-negative bacteria. The objective of this study is to evaluate the ability of lactic acid (LA) to improve PMA penetration and, thus, the efficiency of v-qPCR in detecting the live fraction of pathogens. The pre-treatment of *E. coli* ATCC 8739 cells with 10 mM LA greatly increased PMA penetration into dead cells compared to conventional PMA-qPCR assay, avoiding false positive results. The limit of detection when using LA-PMA qPCR is 1% viable cells in a mixture of dead and alive cells. The optimized LA-PMA qPCR method was reliably able to detect log 2 CFU/mL culturable *E. coli* in milk spiked with viable and non-viable bacteria. Lactic acid is cheap, has low toxicity, and can be used to improve the efficiency of the v-qPCR assay, which is economically interesting for larger-scale pathogen detection applications intended for food matrices.

## 1. Introduction

Pathogen detection is a crucial aspect of maintaining public health, preventing outbreaks, and ensuring the safety of food, soil, water, and other biological samples. According to the World Health Organization, more than 1.5 million individuals worldwide have passed away in the last three years from illnesses brought on by various foodborne infections [1]. *Escherichia coli* is a high-risk food contaminant and can be an indicator of fecal contamination in different foods [2,3]. The conventional method for identifying bacteria is based on culture cultivation, which typically takes two to three days or longer to yield complete results. Culture-based approaches have drawbacks, such as being labor-intensive, lengthy, and susceptible to contamination by germs that are not the target. Both colony-based detection and polymerase chain reaction (PCR) are widespread methods used for the detection of *E. coli* in complex samples such as bodily fluids or food matrices [4,5]. Molecular biology methods, PCR, and quantitative PCR (qPCR) are the most significant contemporary technologies for pathogen detection and identification; however, they cannot discern between living and dead cells, thus leading to false positive results.

Among various molecular biology methods employed for pathogen detection, viable quantitative PCR (v-qPCR) has emerged as a powerful technique that offers high sensitivity and specificity. Viable qPCR combines the principles of qPCR, a sensitive technique for amplifying DNA, with the ability to differentiate between live and dead microorganisms [6,7,8]. Traditional qPCR detects the presence of DNA regardless of whether the microorganism is viable or not. However, viable qPCR goes a step further by incorporating viability dyes, such as propidium monoazide (PMA), ethidium monoazide (EMA), or commercially available reagent D, to selectively target and inhibit the amplification of DNA from non-viable cells while allowing amplification from viable cells [8,9,10]. DNA intercalating dyes are able to enter dead cells without damaging their membrane integrity and intercalate with DNA using their photo-inducible azide groups, preventing PCR-induced DNA amplification [7,8]. Viable qPCR assays with PMA have been successfully optimized to detect major foodborne pathogens, including *E. coli* O157:H7, *Salmonella* spp., and *Listeria monocytogenes* [11,12,13,14,15,16,17].

Despite many successful applications, some practical limitations that impact the accuracy of v-qPCR data have been reported, especially concerning the complexity of the sample matrix, the length of the qPCR amplicon, and the ability of some bacterial strains to avoid dye uptake [18,19]. To increase the PMA uptake without compromising the viability of live cells, different enhancers have been used. Thus, surfactant sodium deoxycholate (DOC) has been used to enhance PMA uptake and improve the efficiency of v-qPCR bacterial detection for *Vibrio vulnificus* killed via freezing or prolonged refrigeration, *E. coli* subjected to mild or pasteurizing heat treatments and heat-killed cells of *Salmonella enterica* serovar *Typhimurium* [20,21,22,23]. However, the co-incubation of Gram-positive *L. monocytogenes* cells with PMA and deoxycholate produces a strong undesired uptake of viability dye by live bacterial cells, suggesting that the effect is strain- and Gram-specific [22]. Additionally, DOC has been proven to enhance the PMA-qPCR results for infectious virus detection in water samples or in a multiplex PMA-PCR developed to detect *Salmonella* spp., *Shigella* spp., and *Staphylococcus aureus* in food products [24,25]. Therefore, other studies used sodium dodecyl sulfate (SDS) to detect viable *Staphylococcus aureus* and *E. coli* in spiked UHT milk samples [26,27]. While maintaining the viability of living cells, SDS has been shown to increase the permeability of dead cells to PMA [28]. Sodium lauroyl sarcosinate (sarkosyl) is a milder detergent than SDS and has proven to be more effective than DOC at boosting PMA signals in the v-qPCR detection of viable *E. coli* from a mixture of alive cells and cells inactivated by heat, lactic acid, or peroxyacetic acid [29]. Recently, SDS-PMA RT-qPCR was developed, a rapid method used to detect SARS-CoV-2 viral particles from biological samples [30]. Similarly, combining PMA and surfactant (sarkosyl 0.025% or triton X-100 0.5%) treatments increased PMA permeability to dead cells and improved v-qPCR detection of *S. aureus* [31]. Moreover, a multiplex PMA-qPCR assay with sarcosyl pre-treatment was successfully applied to detect *Legionella pneumophila*, *S. typhimurium*, and *S. aureus* from water samples [32].

Therefore, this study aimed to contribute to the efforts improving v-qPCR efficiency when testing lactic acid (LA), a new type of PMA-penetrating enhancer. Sublethal concentrations of LA up to 10 mM have been proven to permeabilize the outer membrane of Gram-negative bacteria, causing leakage, damaging the cytoplasmic membrane, and altering the molecular structure [33,34]. Experiments were performed with model Gram-negative bacteria *Escherichia coli* in order to evaluate the ability of LA to improve PMA penetration and, thus, the efficiency of the v-qPCR assay when detecting the alive fraction of bacteria in culture. Moreover, the LA-PMA-qPCR method was applied to the rapid and accurate detection of viable *E. coli* cells in spiked milk.

## 2. Materials and Methods

### 2.1. Bacterial Strain and Culture Conditions

The bacterial strain used in this study, *E. coli* ATCC 8739, was cultured in 10 mL of tryptic soy broth–SB medium (Acumedia, San Bernardino, CA, USA) at 37 °C for 24 h until the stationary phase was achieved. Six and seven serial tenfold dilutions of each culture were prepared and spread onto duplicated tryptic soy agar–TSA media (Scharlau, Spain). The plates were incubated at 37 °C for 24 h, and then the colonies were counted, and the number of viable *E. coli* were determined.

### 2.2. Testing the Sublethal Effect of Lactic Acid on the Strain E. coli ATCC 8739

The sublethal effect of LA was tested using tubes with 3 mL of TSB media with 15 mM, 20 mM, and 30 mM of LA inoculated with 1 mL of overnight *E. coli* culture. After 24 h of incubation at 37 °C, the optical density (OD_600_) was determined. The experiment was performed twice, and the average value was calculated.

### 2.3. Preparation of E. coli Suspensions and Heat Treatment

To obtain dead *E. coli* cells, the overnight cultures were heated at 80 °C for 45 min, and then the loss of cell viability was checked by spreading 1 mL of the heat-treated cell suspensions onto TSA media and incubating the plates for 24 h at an optimal growth temperature. To study signal reduction, aliquots (400 μL/each) of live cells and others with heat-killed cells (400 μL/each) were prepared. The limit of detection study was performed with cell mixtures that contain a defined ratio of viable (100%, 50%, 10%, 1%, 0.1%, and 0%) and dead cells, with a total volume of 400 μL/each.

### 2.4. Pre-Treatment of Cell Suspensions with Lactic Acid

A 30% (*v*/*v*) L-(+)-lactic acid stock solution (Sigma, St Saint Louis, MO, USA) was used to prepare 5 to 30 mM LA solutions with a pH = 5–5.5. Some of the cell aliquots and alive–dead mixtures (400 μL/each) were incubated with 400 μL of LA for 30 min and centrifuged at 150 rpm at room temperature before PMA treatment.

### 2.5. PMA Treatment and Cross-Linking

In this study, 1 mg of PMA (Biotium, Hayward, CA, USA) was dissolved in 98 μL of sterile distilled water to create a stock solution of 20 mM and stored in the dark at −20 °C. Treatment with PMA was performed based on the manufacturer’s instructions, adding 1 μL of stock PMA in 400 μL aliquots to obtain a final concentration of 50 μM. The tubes containing the cell aliquots and mixtures were incubated in the dark at room temperature and centrifuged at 150 rpm for 10 min. Light exposure was performed using a 1000 W halogen light source (Omnilux, Napa, CA, USA) placed 20 cm away from the sample tubes. During the photolysis step, the tubes were kept on ice to avoid excessive heating and occasional mixing. After 5 min of light exposure, cells were harvested via centrifugation (12,000 rpm, for 2 min) and washed twice with sterile distilled water to remove trace amounts of PMA solution that might interfere with free DNA during extraction. The pellet was resuspended in 200 μL of distilled water for DNA isolation.

### 2.6. Extraction of DNA and qPCR

For all samples (LA-PMA-treated, PMA-treated, and non-treated), genomic DNA was extracted from the bacterial cell pellet using a Quick DNA Fungal/Bacterial Miniprep kit (ZymoResearch, Irvine, CA, USA), following the manufacturer’s instructions. Briefly, the resuspended bacterial cells (200 μL) were mixed with 750 μL of BashingBead buffer and mechanically disintegrated using Minibead beater equipment (Biospec Products, Bartlesville, OK, USA) for 2 min. Then, the DNA was isolated after washing. DNA concentration and purity were checked with a QuickDrop Micro-Volume Spectrophotometer (Molecular Devices, San Jose, CA, USA). For all experiments, 1 μL of extracted DNA (up to 50 ng DNA), serving as the template for qPCR assay, was added to 24 μL of a mixture containing 12.5 μL of Maxima SYBR Green/ROX qPCR MasterMix (ThermoScientific, Waltham, MA, USA), 0.5 μL of each primer (forward and reverse primer), and 10.5 μL of PCR-grade water. The following primers were used for the detection of DNA-targeted highly conserved regions of the *uivA-7* gene: forward primer uidA-7-F 5′-GGGATAGTCTGCCAGTTCAGTT-3′ and reverse primer uidA-7-R-deg 5′-GATGTCACDCCGTATGTTATTG-3′ [35]. Quantitative PCR was performed using the Rotor-Gene 6000 5plex HRM (Qiagen-Corbett Life Science, Sydney, Australia) instrument, and software was used to generate the standard curve and for microbial quantification. The 3 steps with the melting program were as follows: denaturation at 95 °C for 10 min, followed by 40 cycles of 15 s denaturation at 95 °C, 30 s annealing at 60 °C, and 20 s elongation at 72 °C, based on the product size (83 bp). In all cases, negative control amplification was included using 1 μL of PCR-grade water (Thermo Fisher Scientific, Waltham, MA, USA) instead of a DNA template. The specificity of qPCR amplification was confirmed by running dissociation curves, and no unspecific products were formed.

For the limit of detection study, the mean of the threshold cycle (Ct) values was used to plot against the natural logarithm of viability (%), and the coefficient of regression (R^2^) values of the corresponding trend line were calculated.

For signal reduction analysis after PMA treatment, delta Ct (dCtPMA) was used for live and dead cells treated or not with PMA, as follows:dC_tALIVE-PMA_ = C_tALIVE, PMA_ − C_tALIVE, non-PMA_
dC_tNON-VIABLE-PMA_ = C_tNON-VIABLE, PMA_ − C_tNON-VIABLE, nonPMA_

For signal reduction analysis after LA-PMA treatments, delta Ct (dCtLA-PMA) was used for live and dead cells treated or not with lactic acid, as follows:dC_tALIVE-LA-PMA_ = C_tALIVE, LA-PMA_ − C_tALIVE, PMA_
dC_tNON-VIABLE-LA-PMA_ = C_tNON-VIABLE, LA-PMA_ − C_tNON-VIABLE, PMA_

The expected results for dC_tALIVE_ should be close to zero (±2), while for dC_tNON-VIABLE_ > 4.

### 2.7. Artificially Inoculated Food Assays

Ultra-high temperature (UHT) sterilized milk was purchased from the local market and used for spiking studies. For standard curve preparation, the overnight culture of *E. coli* ATCC 8739 (1 mL) was inoculated into 1 mL of milk to reach a final concentration of 10^7^ CFU/mL. After the 10 mM LA and PMA treatment, genomic DNA was isolated, and serial dilutions of DNA were used to construct the standard curve. UHT milk was first confirmed to be negative for *E. coli* via plate count on TSA medium and via qPCR analysis. The three milk samples were inoculated with *E. coli* as follows: sample 1—10^2^ CFU/mL alive cells; sample 2—10^2^ CFU/mL dead cells; and sample 3—a mixture of 10^2^ CFU/mL viable and 10^2^ CFU/mL non-viable cells. All of the samples were pre-treated with 10 mM LA, followed by PMA. Then, the genomic DNA was isolated, and qPCR was amplified, as mentioned above.

### 2.8. Statistical Analysis

Statistical analysis was calculated using the IBM SPSS Statistics 23 software package (IBM Corporation, Armonk, NY, USA). The analysis of variance was performed with ANOVA at 95% significance (*p* = 0.05). For qPCR analysis, the average results obtained from two different experiments with triplicates are shown, and the results were compared with ANOVA. The counts obtained after plating were log-transformed, and then the average results and standard deviation were calculated. All of the experiments were performed twice with duplicates or triplicates, and the samples were analyzed within 24 h.

## 3. Results

### 3.1. Effect of LA and PMA Treatments on qPCR Amplification from Viable and Non-Viable Bacterial Cell Suspensions

First, the effect of the PMA treatment on real-time PCR-based detection of culturable and heat-treated *E. coli* ATCC 8739 was evaluated, and the results are shown in Figure 1. Viable bacterial cell suspensions were treated or untreated with PMA prior to DNA isolation and qPCR. The delta C_tALIVE-PMA_ for the viable suspensions is 0.375, a lower value than 2, which proved that PMA did not inhibit the DNA amplification from alive *E. coli* cells. Using suspensions with non-viable cells, treated or untreated with propidium monoazide, the threshold values (C_t_) for non-viable were closer; therefore, the delta C_tNON-VIABLE-PMA_ in this case is 1.025. This result suggests that the viability dye did not efficiently penetrate the outer membrane of *E. coli* and cell wall, and most of the DNA from the dead cells was amplified after DNA isolation. Although the PMA treatment has no adverse effect on the detection of viable *E. coli*, in the case of non-viable bacteria, the high dose of PMA (50 μM) had a low penetration effect. It is considered that a signal reduction higher than 4 for dead cells indicates that 94% of the DNA from these bacteria was removed, and the percentage increases to 99.6% for a value of 8 [36].

Second, the sublethal effect of low concentrations of LA (5–30 mM) on *E. coli* ATCC 8739 cells was evaluated. The permeabilizer function of lactic acid has been known for a long time and explains its antimicrobial effect while supporting its use in decontamination procedures [34]. The viability of *E. coli* cultures that were grown in media with lactic acid up to 20 mM was negligibly affected, while a ~30% decrease in viability was noted in media with 30 mM of LA.

Following previous results relating to the sublethal effect of lactic acid in concentrations up to 20 mM, suspensions of live and heat-killed *E. coli* ATCC 8739 were exposed to different concentrations of LA (5 to 20 mM) and PMA. Control probes were only treated with PMA, followed by photolysis and DNA amplification. Moreover, control probes only treated with LA proved that this compound does not influence DNA amplification. The delta Ct (dC_t_) was defined as the difference in the mean C_t_ values of LA-treated and non-treated probes and reflected the signal reduction level. Increasing concentrations of LA do not significantly influence the delta Ct for viable cells (dC_tVIABLE_), and values lower than 2 were obtained (Figure 1). This proves that viable cells efficiently excluded PMA with or without LA pre-treatment. However, in the case of non-viable 10 mM LA pre-treated cells, the C_tNON-VIABLE, LA-PMA_ signal was 14.5 times higher than the C_tNON-VIABLE, PMA_ obtained without lactic acid incubation. The signal reduction reached the greatest value (dC_tNON-VIABLE_ = 14.5) at a lactic acid concentration of 10 mM before gradually decreasing at higher LA concentrations. Additionally, cells exposed for 30 min to 5 mM and 15 mM lactic acid increased propidium monoazide penetration, and the values for dC_tNON-VIABLE_ were 10.415 and 8.090, respectively (Figure 1).

### 3.2. Limit of Detection for LA-PMA qPCR Assay in a Background of Dead Bacteria

Different works have proved that large quantities of free DNA or DNA from dead cells might interfere with DNA extraction and subsequent qPCR detection from viable cells [8,10,37,38]. Therefore, *E. coli* suspensions with defined ratios of viable (100%, 50%, 10%, 1%, and 0.1%) and non-viable cells were prepared. Half of these suspensions were treated with 10 mM LA and PMA, while the other half were only with PMA, and then the DNA was amplified via real-time PCR with specific primers for the uivA gene. As expected, the highest Ct value was noted for LA-PMA-treated samples with 100% heat-killed cells. A plot of the natural logarithm of DNA percentage versus C_t_ values showed a linear correlation with a coefficient of regression (R^2^) of 0.9537 (Figure 2b) for LA-PMA treated samples and 0.7922 (Figure 2a) for samples exposed to PMA. The improved LA-PMA qPCR assay yielded a good quantification prediction of the viable fraction in dead–alive cell mixtures with higher than 1% viable cells. However, a significant deviation from linearity was noticed in samples with less than 0.1% viable cells.

### 3.3. Detection of E. coli ATCC 8739 via LA-PMA-qPCR Assay in Artificially Inoculated Milk

To evaluate *E. coli* ATCC 8793 detection using LA-PMA-qPCR in food samples, milk samples were artificially spiked with log 2 CFU/mL alive cells, dead cells, and a mix of culturable and heat-killed cells. The standard curve in milk was prepared using 10-fold dilutions of genomic DNA from log 7 CFU/mL *E. coli* culture. The artificially spiked samples and samples for standard curve preparation were pre-treated with LA, followed by PMA and subsequent DNA isolation and qPCR amplification. The correlation coefficient (R^2^ = 0.9792) and the slope (−2.95) of the standard curve were automatically generated, and they showed a good linear relationship between the values of each sample (Figure 3).

The average C_t_ values for alive and mixed alive–dead samples were 25.32 ± 0.30 and 25.69 ± 0.07, respectively, proving that these samples contained log 2 CFU/mL viable bacterial cells. In milk samples inoculated with non-alive *E. coli* cells, the average C_t_ was 28.52 ± 0.55.

## 4. Discussion

Viable qPCR represents a valuable tool in the field of pathogen detection, offering rapid, sensitive, and specific detection of viable pathogens in diverse sample types. Its applications span various fields, including clinical diagnostics, food safety, environmental monitoring, and biodefense, contributing significantly to efforts aimed at controlling infectious diseases and ensuring public health. In recent years, propidium monoazide has been widely used in combination with qPCR assays to limit false positive results in the detection of *Escherichia coli* from different food (e.g., milk, vegetables, and ground meat), environmental, and biological samples [7,15,39,40,41,42]. However, some practical limitations were found, and different strategies were proposed for overcoming some of these problems, including improving the dye penetration, extensive optimization procedures for complex samples, and changing the intercalation dye PMA with a palladium compound [18,19,43]. The above-named limitations led us to seek improved methods to detect and quantify microbes and reduce the false positive/negative results.

To remedy the shortcoming of v-qPCR regarding the ability of PMA dye to penetrate the cells with impaired membrane integrity, a new practical approach based on using lactic acid as an enhancer was proposed. Lactic acid is known as a potent outer membrane-disintegrating agent that causes the release of lipopolysaccharides in Gram-negative bacteria [33,34]. The ability of LA to effectively penetrate the outer membrane of Gram-negative bacteria and act in synergy with antimicrobials has been reported [44]. Recently, the lactic acid cell-penetrating effect has been exploited in various biomedical applications, including the development of lactic acid-based drug delivery systems, gene delivery vectors, and diagnostic probes [45]. Although the mechanisms are still poorly understood, the ability of lactic acid to enhance cellular uptake will improve the targeted delivery of therapeutic agents to specific cells and enhance the efficacy of treatments while minimizing off-target effects [44].

First, we evaluated the ability of LA to improve PMA penetration and, thus, the efficiency of the v-qPCR assay to detect an alive fraction of Gram-negative *E. coli* ATCC 8739. Cell suspensions containing 100% viable bacteria and 100% heat-killed cells were treated or not with different concentrations of LA (5–20 mM), followed by PMA photolysis and DNA amplification. To quantify the signal reduction level, we calculated delta C_t_ (dC_t_) as a difference in the mean C_t_ values of LA-treated and non-treated probes. The mean C_tVIABLE_ values for the DNA extracted from viable cells treated or not treated with LA were not significantly different (*p* > 0.05); thus, the delta C_tVIABLE_ values varied between −0.935 and −1.875. This proved that PMA was not able to enter the viable cells and reduce the C_t_ signal in both cases, with and without LA pre-treatment. However, the mean C_tNON-VIABLE_ values for DNA extracted from killed cells were significantly different (*p* < 0.05), except for bacterial suspensions treated with 20 mM LA and PMA. Based on the dC_tNON-VIABLE_ values, the 10 mM LA pre-treatment had the best effect and improved PMA penetration into heat-killed cells that suppressed the detection of dead cells. Both 5 mM LA and 15 mM LA pre-treatment improved *E. coli* ATCC 8739 PMA-qPCR detection, but the values were lower compared to dC_tNON-VIABLE_ = 14.5 obtained for the 10 mM LA pre-treatment. Similar data were observed by Nkuipou-Kenfack et al., 2013, for the deoxycholate effect on *Salmonella typhimurium* PMA-qPCR detection when PMA penetration decreased at higher concentrations of DOC [22]. In another study concerning *E. coli* detection using SDS-PMA-qPCR, the authors tested different SDS concentrations (0–1000 μg/mL) and reported that the optimal concentration of SDS for enhancing PMA (40 μM) penetration into dead cells was 100 μg/mL [27].

Previous works have shown that the accuracy of PMA-qPCR detection could be affected by the high background of dead cells [8,10,37,38]. To analyze the limit of detection of the LA-PMA-qPCR assay’s defined ratios, culturable viable (100%, 50%, 10%, 1%, and 0.1%) and non-viable cells were prepared. When used for the detection of *E. coli* ATCC 8739 under culture conditions, LA-PMA-qPCR assay gave a good quantitative prediction for viable cells in mixtures of alive–dead cells only when the fraction of viable bacteria was equal to or higher than 1%. Other studies reported that the prediction of *E. coli* and *Listeria innocua* viable cells detected using PMA-qPCR in alive–dead mixtures with less than 1% viable cells was not feasible [21,22]. Testing the effect of PMA treatment on defined ratios of viable (0–100%) and non-viable *E. coli* cells, Nocker et al., 2006, noted a linear correlation R^2^ = 0.9742 after plotting the natural logarithm of normalized DNA concentrations and the corresponding C_t_ values obtained from *stx1* gene amplification [6]. In this case, the PMA-qPCR assay was optimized, and cells were treated with PMA (50 μM) and exposed to light for a longer period of time (120 s), which could explain the differences with other reported data. However, in our experiments, the qPCR assay with the primer pair that targeted the *uivA* gene (amplicon length 83 bp) generated a standard curve with R^2^ = 0.9912 and a slope of −3.45, suggesting robust DNA amplification with these primers. Additionally, it is possible that a longer amplicon of up to 200 bp could increase v-qPCR efficiency, as was recently suggested by Van Holm, 2021 [19].

Finally, LA-PMA-qPCR was successfully applied to detect viable *E. coli* cells in artificially spiked milk. For UHT milk, the standard plating methods require different enrichment media plus long detection/confirmation times (2 days) to target aerobic or anaerobic viable contaminant bacteria. When it comes to analyzing food samples using qPCR, the complexity of food matrices poses several challenges: the presence of PCR inhibitors, variability in food composition, texture, and structure, the presence of enzymes that degrade DNA, and so on [46,47,48,49,50]. In our experiments, the lowest detection limit was log 2 CFU/mL for viable cells or mixtures with log 2 CFU/mL alive and log 2 CFU/mL heat-killed bacteria. Similarly, Dong et al., 2019, reported a detection limit in spiked milk of 3 × 10^2^ CFU/mL viable *E. coli* ATCC 25922 using DOC-PMA-qPCR [27]. Still, other studies have suggested that fat and other components of opaque fluids, such as milk, could protect cells from the effects of DOC and PMA [21,51].

Incorporating lactic acid into viability qPCR assays is relatively straightforward and can be easily integrated into existing laboratory protocols. Moreover, lactic acid is readily available, cost-effective, and compatible with standard laboratory equipment and reagents, making it a practical choice for enhancing the performance of v-qPCR assays. However, more studies are required to evaluate the ability of LA to improve PMA penetration using different species of Gram-positive and Gram-negative bacteria.

## 5. Conclusions

Overall, the results of this study proved that the use of lactic acid in conjunction with viability qPCR offers a powerful strategy for the accurate, sensitive, and specific detection of viable Gram-negative pathogens in a culture and food matrix. Lactic acid (10 mM) greatly enhanced membrane permeability to PMA for heat-killed *E. coli* cells compared to a conventional PMA-qPCR assay (dC_tNON-VIABLE_ = 14.5), thus improving v-qPCR efficiency and the limit of detection in a mixture of dead and alive cells. Lactic acid has low toxicity and is cheap; therefore, it is economically interesting for use in larger-scale applications concerning pathogen detection in food matrices.

By leveraging lactic acid’s ability to enhance and facilitate DNA extraction from viable cells, this approach contributes to advancing pathogen detection methodologies and supporting efforts to safeguard public health and food safety.

## Figures and Tables

**Figure 1 foods-13-01021-f001:**
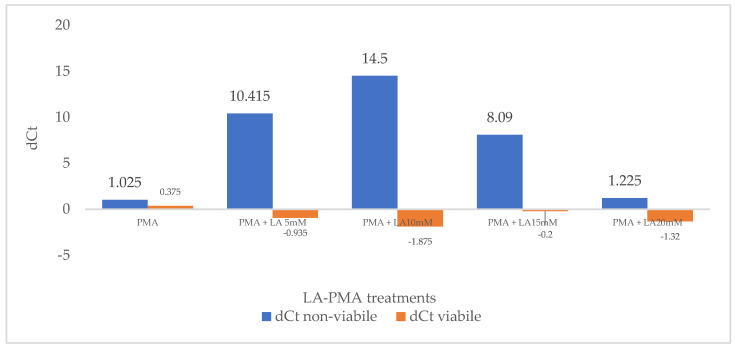
The effect of PMA treatment and LA-PMA treatments on delta dC_t_ for viable and non-viable cell suspensions.

**Figure 2 foods-13-01021-f002:**
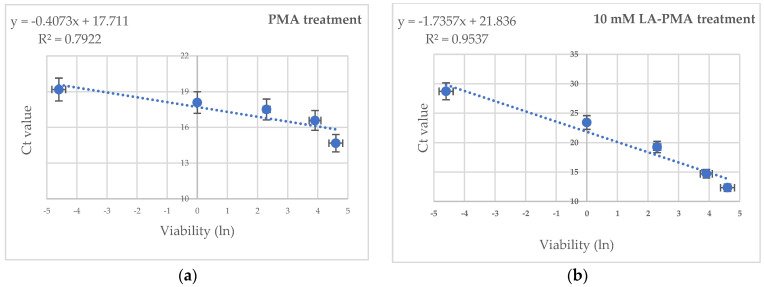
The effect of PMA (**a**) and 10 mM LA-PMA (**b**) treatments on the detection of viable cells in defined ratios of viable (100%, 50%, 10%, and 1%) and dead cells. The average value of the linear coefficients of regression (R^2^) obtained from two independent experiments in triplicate are indicated.

**Figure 3 foods-13-01021-f003:**
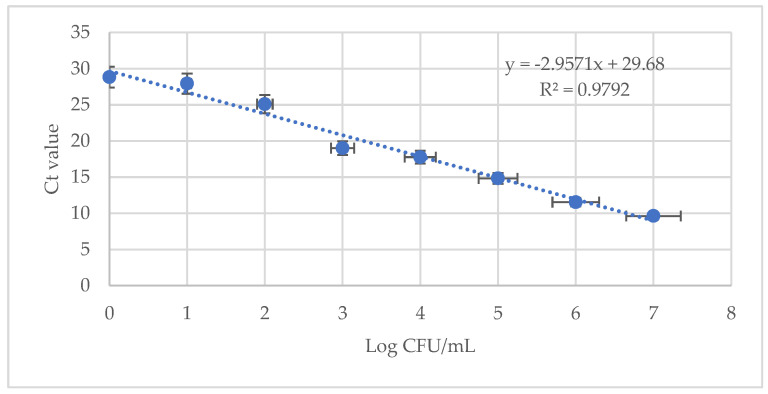
Standard curve for quantification of *E. coli* ATCC 8794 in artificially spiked milk using 10 mM LA-PMA assay. The average value of the linear coefficient of regression (R^2^) obtained in two independent experiments in triplicate is indicated.

## Data Availability

The original contributions presented in the study are included in the article, further inquiries can be directed to the corresponding author.

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
