# Peer review of "Improving the Efficiency of Viability-qPCR with Lactic Acid Enhancer for the Selective Detection of Live Pathogens in Foods"

_foods, 2024, doi:10.3390/foods13071021_

Round 1
Reviewer 1 Report
Comments and Suggestions for Authors
In this paper, the authors report a modification and improvement (more sensitive test) upon the qPCR “traditional” PMA-based method. The method is highly relevant for the routine inspection of complex food matrices that impact public health and food safety. However, here are some points which can be improved before considering the manuscript for publication:
Line 63 Samonella enterica serovar Typhimurium
Line 75 This sentence needs a little rephrasing, it is written awkwardly.
Line 77/79 Please check the correct spelling of the word “sarkosyl”
Line 84 A specific concentration here seems to be unnecessary.
Line 94 Please check “370C” Please also check the degree symbol along the text I suspect that you are using a zero instead of the proper symbol
Line 96 Please check for uniformity in the expression of units there should always be a space between the measurand and the proper unit. Please recheck the whole manuscript for consistency. Check lines 104/117/118 for more examples.
Line 93/96 TSA and TSBA abbreviations should be defined the first time they are used within the text.
Lines 99/100 Please do not use mM as a concentration unit. Please use mmol/L. Please recheck the whole manuscript for consistency. Please also see lines 113/118 for more examples.
Line 104-108 “80 °C” Please revise the use of the letter “l” when describing volumes some are in lower case and some are in upper case. Please recheck the whole manuscript for consistency. Also, see lines 114/117
Line 112 Please reserve the use of the symbol “%” for relative changes. This is not an adequate concentration unit. Please use 30 mL/100 mL. Please recheck the rest of the manuscript.
Line 132 “following the manufacturer’s instructions” is fine as it is an already established method by the supplier. However, a brief description would be appreciated.
Line 162 Please elaborate as to why you chose UHT milk as a model for your food test
Line 174-179 Please describe explicitly which variables you contest during the statistical analysis.
Line 212 Already defined that lactic acid was abbreviated as LA use it along the text.
Line 172 and the rest of the text, figure 1, a letter “d” is not a “delta” please use the correct symbol δ or Δ
Figure 2. Please elaborate as to what the change in slope represents in this case when comparing the LA+PMA vs the PMA alone treatment.
Line 252-253 “samples were 25.32±0.30 and 252 25.69±0.07 respectively” should read “samples were 25.32 ± 0.30 and 252 25.69 ± 0.07 respectively”
Line 273 “Gram-negative” Please recheck the whole manuscript.
The discussion is the one that seems to need a little more attention since I consider it to be a bit brief. I also suggest putting a little more emphasis on the difference that this method makes between viable and non-culturable cells (the differentiation among live/dead cells is clear). Virulence factors and DNA fragments remain even among non-viable and dead cells. The addition of a reference such as this could also help 10.1155/2013/703813. Please include within the discussion which mechanism(s) are expected to be used by the lactic acid, so it improves upon the effect of PMA.
Comments on the Quality of English LanguageEnglish language fine
Author Response
Thank you for the review of our manuscript. We have revised the manuscript in response to comments by the two reviewers. To follow please find an itemized list of each reviewers comments in addition to the manner in which we have addressed the reviewers concern.
In this paper, the authors report a modification and improvement (more sensitive test) upon the qPCR “traditional” PMA-based method. The method is highly relevant for the routine inspection of complex food matrices that impact public health and food safety. However, here are some points which can be improved before considering the manuscript for publication:
Line 63 Samonella enterica serovar Typhimurium
The reviewer is correct and the text has been changed (line 61-65)
Lines 61-65 “Thus, surfactant sodium deoxycholate (DOC) has been used to enhance PMA uptake and improve the efficiency of v-qPCR bacterial detection for Vibrio vulnificus killed by freezing or prolonged refrigeration, E. coli subjected to mild or pasteurizing heat treatments and heat-killed cells of Salmonella enterica serovar Typhimurium [20-23].”
Line 75 This sentence needs a little rephrasing, it is written awkwardly.
The reviewer is correct and the text has been changed (lines 75)
Line 74-77 “Sodium lauroyl sarcosinate (sarkosyl) is a milder detergent than SDS and proved to be more effective than DOC at boosting PMA signals in v-qPCR detection of viable E.coli from a mixture of alive and inactivated by heat, lactic acid or peroxyacetic acid cells [29].”
Line 77/79 Please check the correct spelling of the word “sarkosyl”
Both words “sarkosyl” and “sarcosyl” are correct.
“N-Lauroylsarcosine sodium salt solution
Synonym(s): Sarcosyl, Sarkosyl NL, Sodium N-dodecanoyl-N-methylglycinate, Sodium N-lauroylsarcosinate solution
Linear Formula: CH3(CH2)10CON(CH3)CH2COONa”
CAS No.: 137-16-6”
Line 84 A specific concentration here seems to be unnecessary.
The reviewer is correct and the text has been changed (lines 84)
Lines 85-88 “Sublethal concentrations of LA up to 10 mM have been proven to permeabilize the outer membrane of gram-negative bacteria, causing leakage, damaging the cytoplasmic membrane, and altering the molecular structure [33, 34].”
Line 94 Please check “370C” Please also check the degree symbol along the text I suspect that you are using a zero instead of the proper symbol
The reviewer is correct and the text has been changed (lines 94)
Lines 95-96 “The bacterial strain used in this study E. coli ATCC 8739 was cultured in 10 mL TSB medium (Acumedia, CA, USA), at 370C for 24h until the stationary phase was achieved.“
Line 96 Please check for uniformity in the expression of units there should always be a space between the measurand and the proper unit. Please recheck the whole manuscript for consistency. Check lines 104/117/118 for more examples.
The reviewer is correct and the text has been changed.
Line 93/96 TSA and TSBA abbreviations should be defined the first time they are used within the text.
The reviewer is correct and the text has been changed (lines 93-96).
Lines 95-99 “The bacterial strain used in this study E. coli ATCC 8739 was cultured in 10 mL tryptic soy broth - TSB medium (Acumedia, CA, USA), at 370C for 24 h until the stationary phase was achieved. Six and seven serial tenfold dilutions of each culture were prepared and spread onto duplicated tryptic soy agar - TSA media (Scharlau, Spain).”
Lines 99/100 Please do not use mM as a concentration unit. Please use mmol/L. Please recheck the whole manuscript for consistency. Please also see lines 113/118 for more examples.
Most studies in this field used for permeabiliser mM or M as a concentration unit or % (i.e., (sarkosyl 0.025% or triton X-100 0.5%). It was mention in the text that the stock solution is 30% (v/v) lactic acid and we prepared 5-30 mM solutions, while the LA treatment was performed with 400 μL LA (lines 115-118). However, we changed the text to make it clear:
Lines 115-118 “A 30% (v/v) L-(+)-lactic acid stock solution (Sigma, Missouri, USA) was used to prepare 5 to 30 mM LA solutions, pH = 5 - 5.5. Some of the cell aliquots and alive-dead mixtures (400 μL/each) were incubated with 400 μL LA for 30 min, 150 rpm, at room temperature, before PMA treatment. “
Line 104-108 “80 °C” Please revise the use of the letter “l” when describing volumes some are in lower case and some are in upper case. Please recheck the whole manuscript for consistency. Also, see lines 114/117
The reviewer is correct and the text has been changed to be consistent with the use of letter “L” for volumes.
The reviewer is correct and the text has been changed (lines 104-108)
Lines 107-108 “To obtain dead E. coli cells the overnight cultures were heated at 800C for 45 minutes,…”
Line 112 Please reserve the use of the symbol “%” for relative changes. This is not an adequate concentration unit. Please use 30 mL/100 mL. Please recheck the rest of the manuscript.
In the experiment 2.3 Preparation of E. coli suspension and heat treatment we used a defined ratio of viable (100%, 50%, 10%, 1%, 0.1%, and 0%) and non-viable cells. The 100% shows that all cells in the mixture are viable, while 50% means that half of the cells are viable and half heat-killed, and so on (please, find the below table).
|
Defined ratio |
100% |
50% |
10% |
1% |
0.1% |
0% |
|
Viable (μL) |
400 |
200 |
40 |
4 |
0.4 |
- |
|
Non-viable (μL) |
- |
200 |
360 |
396 |
399.6 |
400 |
|
Total volume (μL) |
400 |
|||||
To make it clear the text has been changed, including “ and 0%”.
Lines 111-114 “The limit of detection study was performed with cell mixtures that contain a defined ratio of viable (100%, 50%, 10%, 1%, 0.1%, and 0%) and dead cells, and a total volume of 400 μl/each.”
Line 132 “following the manufacturer’s instructions” is fine as it is an already established method by the supplier. However, a brief description would be appreciated.
The text has been changed (line 132).
Lines 136-139 “Briefly, the resuspended bacterial cells (200 μl) were mix with 750 μl BashingBead buffer and mechanical disintegrated using Minibead beater equipment (Biospec Products, UK) for 2 minutes, then DNA isolated after washing. “
Line 162 Please elaborate as to why you chose UHT milk as a model for your food test
The reviewer has raised a good point. Long detection/confirmation times of contaminants and the presence of lipids in the milk that were reported to limit the PMA penetration are the main reasons why we tested the new LA-PMA-qPCR method using this model. Therefore, the text has been changed to make clear this aspect.
Lines 335-346 “Finally, the LA-PMA-qPCR was successfully applied to detect viable E. coli cells in artificially spiked milk. For UHT milk the standard plating methods require different enrichment media plus long detection/confirmation times (2 days) to target aerobic or anaerobic viable contaminant bacteria. When it comes to analyzing food samples using qPCR, the complexity of food matrices poses several challenges: the presence of PCR inhibitors, variability in food composition, texture, and structure, the presence of enzymes that degrade DNA, and so on [46-50]. In our experiments, the lowest detection limit was log 2 CFU/mL for viable cells or mixture with log 2 CFU/ mL alive and log 2 CFU/mL heat-killed bacteria. Similarly, Dong et al., 2019 reported a detection limit in spiked milk of 3 x 102 CFU/mL viable E. coli ATCC 25922 using DOC-PMA-qPCR [27]. Still, other studies suggested that fat and other components of opaque fluids, such as milk could protect cells from the effect of DOC and PMA [21, 51]. “
Line 174-179 Please describe explicitly which variables you contest during the statistical analysis.
The text has been changed:
Lines 180-186 “The statistical analysis was calculated using the IBM SPSS Statistics 23 software package (IBM Corporation, USA). The analysis of variance was performed with ANOVA at 95% significance (P=0.05). For qPCR analysis, the average results obtained from two different experiments with triplicates are shown and results were compared with ANOVA. The counts obtained after plating were log transformed, then average results and standard deviation were calculated. All experiments were performed twice with duplicates or triplicates and the samples were analyzed within 24 hours.”
Line 212 Already defined that lactic acid was abbreviated as LA use it along the text.
The reviewer is correct and the text has been changed (line 212)
Line 214-215 “Increasing concentrations of LA do not significantly influence the delta Ct for viable cells (dCtVIABLE) and values lower than 2 were obtained (Figure 1).“
Line 172 and the rest of the text, figure 1, a letter “d” is not a “delta” please use the correct symbol δ or Δ
The reviewer is correct and symbol for “delta is δ or Δ. However, the manufacturer of PMA (Biotium, USA) use “dCt” in the Product Information flyer and explained how it could be calculated and the expected values for dCtVIABLE and dCtNON-VIABLE. Therefore, we would like to keep this abbreviation to avoid any misunderstanding.
Figure 2. Please elaborate as to what the change in slope represents in this case when comparing the LA+PMA vs the PMA alone treatment.
The reviewer has raised a good point. However, it was mentioned in the text that LA-PMA assay with R2=0.95 gave a better prediction of the viable fraction in mixtures with alive and dead cells.
Lines 239-243 “The improved LA-PMA qPCR assay gave a good quantification prediction of the viable fraction in dead-alive cell mixtures with higher than 1% viable cells. However, a significant deviation from linearity was noticed in samples with less than 0.1% viable cells.”
Line 252-253 “samples were 25.32±0.30 and 252 25.69±0.07 respectively” should read “samples were 25.32 ± 0.30 and 252 25.69 ± 0.07 respectively”
The text has been changed (lines 252-253)
Lines 262-265 “The average Ct values for alive and mix alive-dead samples were 25.32 ± 0.30 and 25.69 ± 0.07 respectively, proving that…”
Line 273 “Gram-negative” Please recheck the whole manuscript.
The reviewer is correct and the text has been changed.
The discussion is the one that seems to need a little more attention since I consider it to be a bit brief. I also suggest putting a little more emphasis on the difference that this method makes between viable and non-culturable cells (the differentiation among live/dead cells is clear). Virulence factors and DNA fragments remain even among non-viable and dead cells.
The reviewer has raised a good point. Howevere, it was mentioned that PMA prevent PCR amplification from dead cells (lines 52-54). In the last 20 years many studies and reviews focused on the importance of this method, including their ability to difference dead and viable cells and that was mentioned in the text (lines 267-274).
Lines 52-54 “DNA intercalating dyes are able to enter dead cells with damage membrane integrity and intercalate with DNA using their photo inducible azide groups, preventing PCR-induced DNA amplification [7, 8].
Lines 267-274 “Viable qPCR represents a valuable tool in the field of pathogen detection, offering rapid, sensitive, and specific detection of viable pathogens in diverse sample types. Its applications span various fields including clinical diagnostics, food safety, environmental monitoring, and biodefense, contributing significantly to efforts aimed at controlling infectious diseases and ensuring public health. In the last years, propidium monoazide has been widely used in combination with qPCR assays to limit false-positive results in the detection of Escherichia coli from different food (e.g., milk, vegetables, ground meat), environmental and biological samples [7, 15, 39-42].”
The addition of a reference such as this could also help 10.1155/2013/703813. Please include within the discussion which mechanism(s) are expected to be used by the lactic acid, so it improves upon the effect of PMA.
The reviewer has raised a good point. However, we already mentioned that the LA mechanism is poorly understood (line 289-290). To make it clear the text has been changed
Lines 283-285 “Lactic acid is known as a potent outer membrane-disintegrating agent that cause lipopolysaccharide release for Gram-negative bacteria [33, 34]. The ability of LA to effectively penetrate the outer membrane of Gram-negative bacteria and act in synergy with antimicrobials has been reported [44]. Recently, the lactic acid cell-penetrating effect has been exploited in various biomedical applications, including the development of lactic acid-based drug delivery systems, gene delivery vectors, and diagnostic probes [45]. Although the mechanisms are still poorly understood, the ability of lactic acid to enhance cellular uptake will improve the targeted delivery of therapeutic agents to specific cells, and enhance the efficacy of treatments, while minimizing off-target effects [44].”
Reviewer 2 Report
Comments and Suggestions for Authors
Nice work with good analytical conclusion, valuable for the readers: Improving the efficiency of viability-qPCR with lactic acid enhancer for selective detection of live pathogens in foods by Laura-Dorina Dinu et al…
Abstract or conclusion: “Lactic acid greatly enhanced membrane permeability to PMA for heat-killed E. coli cells, compared to conventional PMA-qPCR assay (dCtNON-VIABLE=14.5) and thus improving the v-qPCR efficiency and limit of detection in a mixture of dead and alive cells”
I would put this number 0f 14.5 into 2^14.5 scale (more then 10 000 times) so reader not familiar with Ct unit scale can recognize the effect.
Lane 29, missing “per unit of time” (year)?
Missing turnaround time estimation for the moment sample arrives to the result?
I would be critical for the future problems, challenges from the industry:
1) indicating that applicability on gram negative “only” method is making food safety screening complex.
2) different effect on different species
3) the other options of “sterilization”?
4) the products which have probiotics but not. ..(species identification)
One can list examples related to industrial/agricultural safety issues where these type of testing will help.
Author Response
Thank you for the review of our manuscript. We have revised the manuscript in response to comments by the reviewers. To follow please find an itemized list of each reviewer's comments in addition to the manner in which we have addressed the reviewer concern.
Nice work with good analytical conclusion, valuable for the readers: Improving the efficiency of viability-qPCR with lactic acid enhancer for selective detection of live pathogens in foods by Laura-Dorina Dinu et al…
Abstract or conclusion: “Lactic acid greatly enhanced membrane permeability to PMA for heat-killed E. coli cells, compared to conventional PMA-qPCR assay (dCtNON-VIABLE=14.5) and thus improving the v-qPCR efficiency and limit of detection in a mixture of dead and alive cells”
I would put this number 0f 14.5 into 2^14.5 scale (more then 10 000 times) so reader not familiar with Ct unit scale can recognize the effect.
The reviewer is correct and the text has been changed.
Lines 17-19 “Pre-treatment of E.coli ATCC 8739 cells with 10 mM LA greatly increased PMA penetration into dead cells, compared to conventional PMA-qPCR assay, avoiding the false-positive results. “
Lane 29, missing “per unit of time” (year)?
The reviewer is correct and the text has been changed (line 29).
Lines 30-32 “According to the World Health Organization, more than 1.5 million individuals worldwide passed away in the last three years from illnesses brought on by various foodborne infections [1].”
Missing turnaround time estimation for the moment sample arrives to the result?
The reviewer is correct and the text has been changed.
Lines 184-186 “All experiments were performed twice with duplicates or triplicates and the samples were analyzed within 24 hours.”
I would be critical for the future problems, challenges from the industry:
1) indicating that applicability on gram negative “only” method is making food safety screening complex.
2) different effect on different species
3) the other options of “sterilization”?
4) the products which have probiotics but not. ..(species identification)
The reviewer has raised a good point and the text has been changed:
Lines 351-352 “ However, more studies are required to evaluate the ability of LA to improve the PMA penetration using different species of Gram-positive and Gram-negative bacteria “
One can list examples related to industrial/agricultural safety issues where these type of testing will help.
The reviewer is correct but we already mentioned the (LA) v-qPCR applications and advantages /limitations.
Lines 268-270 “ Its applications span various fields including clinical diagnostics, food safety, environmental monitoring, and biodefense, contributing significantly to efforts aimed at controlling infectious diseases and ensuring public health.”
Lines 277-279 “The above-named limitations led us to seek improved methods to detect and quantify microbes and reduce the false positive/negative results. ”
Lines 347-351 “Incorporating lactic acid into viability qPCR assays is relatively straightforward and can be easily integrated into existing laboratory protocols. Moreover, lactic acid is readily available, cost-effective, and compatible with standard laboratory equipment and reagents, making it a practical choice for enhancing the performance of v-qPCR assays. ”
Reviewer 3 Report
Comments and Suggestions for Authors
The manuscript (foods-2908295) provided an optimized LA-PMA qPCR method which can fulfill the detection of log 2 CFU/mL culturable E.coli in milk spiked with viable and non-viable bacteria. This method has scientific significance and economical interests. However, there are some parts of experiments should be reorganized or improved. As a result, I give the suggestion of accepting this manuscript after major revision.
Here are some questions and suggestions:
1. It is common agreement that biological experiments need to be repeated at least three times. However, in Figure 1, we can see no trace of experimental repetition or statistical calculations performed.
2. Figure 1 should be made more clear to give the reader a better reading experience. For example, the X-axis title named PMA+LA 15mM coincides with -0.2.
3. Authors must give a clear expression of all results. I think the expression should be improved. For example, the meaning of the result in Figure 3 was not expressed clearly.
Author Response
Thank you for the review of our manuscript. We have revised the manuscript in response to comments by the reviewers. To follow please find an itemized list of each reviewers comments in addition to the manner in which we have addressed the reviewer concern.
The manuscript (foods-2908295) provided an optimized LA-PMA qPCR method which can fulfill the detection of log 2 CFU/mL culturable E.coli in milk spiked with viable and non-viable bacteria. This method has scientific significance and economical interests. However, there are some parts of experiments should be reorganized or improved. As a result, I give the suggestion of accepting this manuscript after major revision.
Here are some questions and suggestions:
- It is common agreement that biological experiments need to be repeated at least three times. However, in Figure 1, we can see no trace of experimental repetition or statistical calculations performed.
The reviewer has raised a good point. However, the delta Ct (dCt) was defined as the difference between two numbers (average values without standard deviation-SD). Figures 2 and 3 showed average values and SD. Similar, in the experiment with artificially inoculated milk the mean Ct values with standard deviation were mentioned (lines 262-265)
Lines 262-265 “The average Ct values for alive and mix alive-dead samples were 25.32 ± 0.30 and 25.69 ± 0.07 respectively, proving that these samples contained log 2 CFU/mL viable bacterial cells. In milk samples inoculated with non-alive E. coli cells the average Ct was 28.52 ± 0.55. “
- Figure 1 should be made more clear to give the reader a better reading experience. For example, the X-axis title named PMA+LA 15mM coincides with -0.2.
The negative values for dCt is related to the definition of the delta Ct (dCt). This is a difference between two average numbers and the result can be positive (i.e., 0.375) or negative (-1.875), but lower than 2 (lines 215-217).
The X-axis title is “LA-PMA treatment” and Y-axis title is dCt. The X-title was moved down.
Lines 215-217 “Increasing concentrations of LA do not significantly influence the delta Ct for viable cells (dCtVIABLE) and values lower than 2 were obtained (Figure 1). This is a prove that viable cells efficiently excluded PMA with or without LA pre-treatment. “
- Authors must give a clear expression of all results. I think the expression should be improved. For example, the meaning of the result in Figure 3 was not expressed clearly.
The reviewer is correct and the text has been changed.
Lines 254-257 “The correlation coefficient (R2 = 0.9792) and the slope (-2.95) of the standard curve were automatically generated and showed that there is a good linear relationship between the values of each sample (Figure 3). ”
Lines 335-346 “Finally, the LA-PMA-qPCR was successfully applied to detect viable E. coli cells in artificially spiked milk. For UHT milk the standard plating methods require different enrichment media plus long detection/confirmation times (2 days) to target aerobic or anaerobic viable contaminant bacteria. When it comes to analyzing food samples using qPCR, the complexity of food matrices poses several challenges: the presence of PCR inhibitors, variability in food composition, texture, and structure, the presence of enzymes that degrade DNA, and so on [46-50]. In our experiments, the lowest detection limit was log 2 CFU/mL for viable cells or mixture with log 2 CFU/ mL alive and log 2 CFU/mL heat-killed bacteria. Similarly, Dong et al., 2019 reported a detection limit in spiked milk of 3 x 102 CFU/mL viable E. coli ATCC 25922 using DOC-PMA-qPCR [27]. Still, other studies suggested that fat and other components of opaque fluids, such as milk could protect cells from the effect of DOC and PMA [21, 51]. ”
Round 2
Reviewer 3 Report
Comments and Suggestions for Authors
The manuscript (foods-2908295) has been improved. However, the issues raised last time have not been fully addressed. As a result, I give the suggestion of accepting this manuscript after minor revision.
Here are some questions and suggestions:
1. The author has misunderstood the question I asked. In Figure1, for the sake of good reading experience, it is recommended not to overlap any labels, such as "PMA+LA 15mM" and "-0.2".
Author Response
Reviewer #3
The author has misunderstood the question I asked. In Figure1, for the sake of good reading experience, it is recommended not to overlap any labels, such as "PMA+LA 15mM" and "-0.2".
Sorry for that misunderstanding. Figure 1 has been changed.